# Nitrogen Balance of Dairy Cows Divergent for Milk Urea Nitrogen Breeding Values Consuming Either Plantain or Perennial Ryegrass

**DOI:** 10.3390/ani11082464

**Published:** 2021-08-22

**Authors:** Cameron J. Marshall, Matthew R. Beck, Konagh Garrett, Graham K. Barrell, Omar Al-Marashdeh, Pablo Gregorini

**Affiliations:** 1Faculty of Agriculture and Life Sciences, Lincoln University, P.O. Box 85084, Lincoln 7647, New Zealand; Konagh.Garrett@lincolnuni.ac.nz (K.G.); Graham.barrell@lincoln.ac.nz (G.K.B.); omar.al-marashdeh@lincoln.ac.nz (O.A.-M.); pablo.gregorini@lincoln.ac.nz (P.G.); 2USDA-ARS, Livestock Nutrient Management Research Unit, 300 Simmons Drive, Unit 10, Bushland, TX 79012, USA; matthew.r.beck@usda.gov

**Keywords:** N excretion, environmental impact, plantain, ryegrass, dairy cows

## Abstract

**Simple Summary:**

We studied the nitrogen excretion patterns of cows selected for divergent nitrogen excretion consuming either ryegrass or plantain. Both the use of a plantain diet as well as the use of low milk urea nitrogen breeding values were found to reduce the concentration of urinary urea nitrogen per urination event compared to cows with a high milk urea nitrogen breeding value and cows consuming a ryegrass-based diet. These results indicate that both the use of cows with low milk urea nitrogen breeding values and the use of a plantain diet are tools that temperate pastoral dairy production systems can use to reduce nitrogen loses.

**Abstract:**

Inefficient nitrogen (N) use from pastoral dairy production systems has resulted in environmental degradation, as a result of excessive concentrations of urinary N excretion leaching into waterways and N_2_O emissions from urination events into the atmosphere. The objectives of this study were to measure and evaluate the total N balance of lactating dairy cows selected for milk urea N concentration breeding values (MUNBVs) consuming either a 100% perennial ryegrass (*Lolium perenne* L.) or 100% plantain (*P**lantago lanceolata* L.) diet. Sixteen multiparous lactating Holstein-Friesian × Jersey cows divergent for MUNBV were housed in metabolism crates for 72 h, where intake and excretions were collected and measured. No effect of MUNBV was detected for total N excretion; however, different excretion characteristics were detected, per urination event. Low MUNBV cows had a 28% reduction in the concentration of urinary urea nitrogen (g/event) compared to high MUNBV cows when consuming a ryegrass diet. Cows consuming plantain regardless of their MUNBV value had a 62% and 48% reduction in urinary urea nitrogen (g/event) compared to high and low MUNBV cows consuming ryegrass, respectively. Cows consuming plantain also partitioned more N into faeces. These results suggest that breeding for low MUNBV cows on ryegrass diets and the use of a plantain diet will reduce urinary urea nitrogen loading rates and therefore estimated nitrate leaching values, thus reducing the environmental impact of pastoral dairy production systems.

## 1. Introduction

Nitrate leaching is a key environmental concern from temperate pastoral dairy production systems, largely as a result of an oversupply of dietary nitrogen (N) [1]. Dairy cows typically only utilise 30% of dietary N for milk production, with approximately 70% of ingested N being excreted as a waste product [2]. Over 60% of this surplus N is excreted as urinary N (UN) [3,4], which is in an immediately available form for leaching or volatilisation [1]. Approximately 82% of UN is discharged to pastures in temperate pasture-based systems [5,6], with 20–30% of this UN being leached to groundwater as nitrates [7] and 2% transformed to nitrous oxide (N_2_O) [8]. Excessive concentrations of N in waterways are linked to environmental degradation, such as eutrophication of waterways, while N_2_O is a potent greenhouse gas (GHG) [9]. This environmental degradation reduces the ability of countries where temperate pastoral dairy production practises are used, such as New Zealand, to meet their commitments under global conventions, such as the Kyoto protocol and Ramsar convention [10], and reduces their ability to meet ethical [11] and sustainable development goals set out by the United Nations [12]. There has also been an association between high concentrations of nitrates in drinking water and several negative human health outcomes, such as increased risk of colorectal cancer [13], methemoglobinemia in infants [14], and neural tube defects [15].

Dietary and management strategies focusing on diverse swards and grazing management have been extensively explored as ways to reduce N losses [4,16,17,18,19], with the use of narrow-leaf plantain (*Plantago lanceolata* L. (PL)) showing promising results for reducing total N excretions and UN concentrations in dairy cows [20,21]. PL is known to contain secondary bioactive compounds, such as acubin, catalpol, and acteoside [22,23], that have been reported to have antimicrobial and diuretic effects [22,24], which may influence the N partitioning of the animals. The use of animal genetics to reduce environmental impact is being explored. Several studies have proposed the potential of using milk urea nitrogen concentration (MUN) values as a predictor for urinary urea excretions, as urea is known to equilibrate between the body fluids of an animal [25], with a linear relationship between the blood and urinary urea pools being established [26,27,28]. The heritability of MUN has been well documented, with estimates ranging between 0.14 and 0.24 [29,30,31]. Beatson et al. (2019) [31] proposed that breeding for lower MUN may present an opportunity to reduce urinary N excretion. Marshall et al. (2020) [32] demonstrated a relationship between MUN breeding values (MUNBVs) and urinary urea nitrogen (UUN) excretion, showing that in a grazing setting, dairy cows with a low MUNBV had reductions in UUN excretion, thus indicating a potential for animal-based solutions to reduce nitrogen losses.

While the relationship between MUNBV and UUN excretion has been demonstrated in a grazing environment, to our knowledge, this relationship has not been conducted in a controlled environment with all urination, defecation, and milk production events being collected and collated into a total daily N balance. Any interaction between diet and MUNBV is also yet to be reported.

We hypothesized that cows divergent for MUNBVs will have differences in their nitrogen balance, with the effect of diet being additive to MUNBV for reducing UUN excretion. The objectives of this study were, therefore, to capture and collate individual intake, urination, defecation, and milk production events into a total N balance of lactating dairy cows divergent for MUNBV; and secondly, to evaluate the effect of a monoculture diet of PL compared with a monoculture diet of perennial ryegrass (*Lolium perenne* L. (RG)) diet on relationships between MUNBVs and milk production and excretion parameters.

## 2. Materials and Methods

This study was conducted at Lincoln University’s Johnstone Memorial Laboratory in Canterbury, New Zealand (43°64′ S, 172°45′ E) during January and February 2020. Sixteen multiparous lactating Holstein-Friesian × Jersey cows (150.4 ± 20.7 days in milk; 506 ± 35 kg live weight; 3.75 ± 0.25 body condition score; 6.7 ± 1.4 years old; 4.6 ± 1.4 lactations) divergent for their estimated MUNBV mg/dL were used. Milk urea nitrogen breeding values were estimated as described by Beatson et al. (2019) [31] and were provided as part of a regular commercial herd test conducted by CRV (New Zealand). Cows were selected based on their level of divergence from the herd’s mean value: the cows used during this study had an MUNBV range from −1.4 to +3.3.

A 2 × 2 factorial design with repeated measures was implemented over 4 runs. Each run consisted of 12 days (9 acclimation days, 3 measurement days in metabolism crates). Each run was conducted concurrently with a 3-day delay so that as the first group of cows left the metabolism crates, the next group entered. During each run, cows were grouped as either high or low for MUNBV (8 high vs. 8 low) and diet type (RG (cv. One50) or PL (cv. Agritonic)). Four cows classified as high for MUNBV and 4 cows classified as low for MUNBV were allocated to each of the 2 dietary treatments (PL or RG), resulting in 1 animal per factorial arrangement (e.g., 1 high PL, 1 high RG, 1 low PL, 1 low RG) per run. The average MUNBV for cows classified as ‘high’ in the PL treatment was 2.19 ± 0.21 and −1.23 ± 0.21 for cows classified as low. Cows in the RG treatment had an average MUNBV of 2.23 ± 0.21 for cows classified as high and −1.09 ± 0.21 for those classified as low. Cows were balanced for production worth, age, and live weight across both dietary treatments and MUNBV groups.

All cows grazed a ryegrass-based sward before the start of the experiment and were all dosed with Rumensin capsules (Elanco, Auckland, New Zealand) 1 week before the start of experimentation to help prevent bloat. A nine-day indoor acclimation period was used to acclimate cows to their allocated diets and feeding bins before cows were housed in metabolism crates for 72 h. The plantain proportion in the diet was increased by 20% each day starting at a baseline level of 20% during the acclimation period, with the remaining proportion of the diet being ryegrass. In this way, cows were on a 100% diet of plantain by day 5. Cows had *ad libitum* access to fresh water during both the acclimation and measurement periods. During the measurement period, PIUSI K24 (PIUSI, Via pacinotti, Italy) flow rate meters were installed in the troughs in the metabolism crates to measure water intake.

### 2.1. Herbage Measurements

Fresh herbage was cut daily using a Haldrup (GmbH, Ilshofen, Germany) forage harvester at a height of 3 cm from a ~30 cm height standing sward before being stored in a chiller at 4 °C and fed non-chopped within 24 h of harvest. Twice daily, after morning and afternoon milking (0700 and 1500 h, respectively), refusals from the previous period were removed and weighed before new herbage was weighed and allocated to the cows. A 10% refusal of as fed dry matter (DM) per feeding for each cow was targeted to provide *ad libitum* access to feed. At each feeding, samples of allocated and refused herbages were taken for analysis of DM content, and chemical and botanical composition.

Botanical composition was determined for both allocated and refused feed by splitting a representative sample into its constituent parts before being oven-dried for 72 h (60 °C). The chemical composition of both allocated and refused feed was determined for each feeding by taking a representative sample from each dietary treatment. Subsamples were lyophilized and ground using a 1-mm sieve in a centrifugal mill (ZM200 Retsch, Haan, Germany) and analysed for quality attributes. Near-infrared spectrophotometry (NIRS, Model: FOSS NIRSystems 5000, Laurel, MD, USA) was used to analyse organic matter (OM), water-soluble carbohydrates (WSCs), neutral and acid detergent fibre (NDF, ADF), crude protein (CP), and DM, OM, and dry OM digestibility (DMD, OMD, DOMD, respectively). The metabolizable energy (ME) content of forage was calculated as per the calculation of the primary Industries Standing Committee (2007) [33] of 0.16 × DOMD. NIRS calibration equations were determined prior to sample analysis for OM by digestibility (R^2^ = 0.89) [34], WSC (R^2^ = 0.97) [35], ADF (R^2^ = 0.97) (AOAC, 1990 [36]; method 973.18), NDF (R^2^ = 0.98) [37], CP by combustion (R^2^ = 0.99) (Vario Max CN Analyser Elementar, Langenselbold, Germany), DMD (R^2^ = 0.97), OMD (R^2^ = 0.97), and DOMD (R^2^ = 0.92) [34]. 

### 2.2. Animal Measurements

Cows were monitored for the entire duration that they were housed in the metabolism crates by trained technicians who were unaware of each animal’s MUNBV. Cows were fitted with a 3-D-printed attachment constructed of acrylonitrile butadiene styrene printed using an UP-Mini 3-D printer (3D Printing Systems, Auckland, NZ), which fitted over the vulva of the animal and was secured with the use of a biocompatible glue (Loctite 454 was used; Henkel, Düsseldorf, Germany). Attached to the 3-D-printed attachment was approximately 1 m of plastic vinyl sleeve, which ‘hung free’ behind the animal and allowed the flow of a urination event to be diverted into a bucket by a technician. In this manner, the urination pattern was able to be recorded per event with every event being individually captured in a bucket, with the time and weight recorded. A subsample of each urination event was immediately acidified with 0.1 mL of sulfuric acid (concentration ≥95%) from a 1-mL transfer pipette and capped to minimise ammonia (NH_3_) volatilisation before being frozen. Total collection trays were in place to capture any urine that may have been missed or spilt. These trays were emptied and weighed at both morning and afternoon milkings. Urine from the total collection trays was not used for chemical analysis due to the possibility of faecal contamination. The volume of urine missed from the bucket was calculated to be consistent per event, with the weight of the urine in the tray being divided by the number of events that occurred during these periods and added to the volume of each event to create an adjusted weight. On average, 30 mL were missed per urination event. A commercial enzymatic kinetic technique using a clinical analyser (Randox Daytona; Crumlin, UK) was used to determine UUN concentration. Urine total N content was determined by combustion (Vario Max CN Analyser Elementar, Langenselbold, Germany) from samples that were composited by volume depending on the relative urination event size from each 24-h period that the cows were housed in the metabolism crates.

Nitrogen deposited to pasture was calculated as the N loading rate using Equation (1) of Haynes and Williams (1993) [38] as previously implemented by Marshall et al. (2020) [32]. The loading rate of UUN per urine patch can be estimated as:(1)$$\mathrm{UUN}\;\mathrm{rate}\;\left({\mathrm{kg}\;\mathrm{UUN}\;\mathrm{ha}}^{-1}\right)=\mathrm{Concentration}\;\left({g\;\mathrm{UUN}\;L}^{-1}\right)\times \frac{\mathrm{Vol}\;\left(L\right)}{\mathrm{Area}\;\mathrm{per}\;\mathrm{urination}\;\mathrm{event}\;\left(m^{2}\right)}\times 10$$

The area covered by the urine patch was estimated using the following equation of Romera et al. (2012) [39] (2) assuming a urine column size of 5 mm:(2)$$\mathrm{Area}\;\mathrm{per}\;\mathrm{urination}\;\mathrm{event}=\frac{\mathrm{Average}\;\mathrm{volume}\;\mathrm{per}\;\mathrm{event}\;\times \frac{1000}{10,000}}{\mathrm{Urine}\;\mathrm{column}\times 0.1}$$

Purine derivatives were determined from an average weighted urine sample for each 24-h period cows were housed in the metabolism creates. A modified HPLC methodology [40] was used on an Agilent 1100 series chromatograph (Agilent Technologies, Santa Clara, CA, USA) with a prodigy 250 × 4.6 mm, 5 u, ODS, C18 reverse-phase column (phenomenex, Torrance, CA, USA). Total purine excretion was calculated as the sum of allantoin and uric acid [41] multiplied by the total volume of urine per day. Xanthine and hypoxanthine values were negligible and therefore not included in total purine derivatives; these purines are often not detected in cattle urine [42,43]. Total purine absorbed was then calculated using the equation of International Atomic Energy Agency (1997) [41] (3):(3)$$y=0.85x+\left(0.385\;LW^{0.75}\right)$$
where y = PD excretion and x = purine absorption (mmol/d) and LW ^0.75^ = metabolic liveweight of the animal (kg).

Microbial CP supply was then estimated using the following Equation (4), where χ = purine absorbed:(4)$$MicrobialN\left(gN/d\right)=\frac{\chi \left(mmol/d\right)\times 70}{0.116\times 0.83\times 1000}=0.727\chi$$

The N content of purines was assumed to be 70 mg N/mmol, the ratio of purine N to total N in mixed rumen microbes was 11.6:100, and the digestibility of microbial purines was assumed to be 0.83 [41]. Microbial CP was then calculated by multiplying microbial N by 6.25. Microbial synthesis efficiency was calculated as grams of urinary microbial CP per kilogram of truly digestible OM intake [42,44].

Faecal outputs were recorded in the same manner as urine outputs, with each event being collected in a bucket by a technician with the weight and time being recorded. A subsample was taken per event and frozen for analysis. A screen above the urine total collection tray was in place to capture any missed faecal material per event. Technicians were able to collect all faecal material per event as any missed material was collected from the screen. Faecal samples were lyophilized to determine faecal DM before being ground through a 1-mm sieve (ZM200 Retsch, Haan, Germany) and analysed for OM, NDF, ADF, and N percentage by NIRS (Model: FOSS NIRSystems 5000, Laurel, MD, USA). NIRS calibration equations were determined for faeces prior to sample analysis for OM by digestibility (R^2^ = 0.93) [34], ADF (R^2^ = 0.99) (AOAC [36], 1990; method 973.18), NDF (R^2^ = 0.99) [37], N% by combustion (R^2^ = 0.98) (Vario Max CN Analyser Elementar, Langenselbold, Germany), and DMD (R^2^ = 0.96) [34].

Cows were milked twice daily at 07:00 and 15:00 h throughout the adaptation and measurement period. A portable milking machine (DeLaval vacuum pump DVP170; DeLaval: Tumba, Sweden) consisting of two separate lines and cisterns was used so that each cow’s milk could be kept separate, with samples collected during the 3 days of the measurement period assessed for yield and quality attributes. Lines were flushed with 20 L of cold water and ‘sucked’ dry between cows to avoid any cross-contamination of milk between cows. Representative 100-mL milk samples were taken per milking and analysed by MilkTest, New Zealand using a CombiFoss machine (Foss Electric, Hillerød, Denmark) for quality attributes (protein, fat, lactose, and somatic cell count) and MUN was measured on skimmed milk by an automated Modular P analyser (Roche/Hitachi; [45]). Milk DM was determined after drying a sample for 5 days at 60 °C. 

Rumen samples were collected from each animal by oesophageal tubation on the first day of the adaption period and the last day of the measurement period (days 1 and 12). Volatile fatty acid (VFA) concentrations were determined by gas chromatography [46] using a gas chromatographer (GC-2010, Shimadzu, Kyoto, Japan) fitted with an SGE BP21 30 m × 530 μm × 1.0 μm bore capillary column, VFAs are grouped and reported as glucogenic and non-glucogenic VFAs. NH_3_ concentration was determined on a rumen fluid sub-sample acidified using sulphuric acid by an enzymatic UV method [47], using an automated clinical analyser (Randox Rx Daytona, Crumlin, UK).

### 2.3. Statistical Analysis 

Data were tested for normality using Shapiro–Wilk’s test to satisfy the assumption of normality for the statistical models. Data that were normally distributed were analysed using a mixed model ANOVA using the ‘lme4′ package, and a type III analysis of variance table using Satterthwaite’s method for determining degrees of freedom was used to determine *p* values for fixed effects and interactions. Non-normally distributed data were analysed using a generalised linear mixed model ANOVA using the ‘lme4’ package, and a type II analysis of variance table using Wald chi-square method was used for determining *p* values for fixed effects and interactions. Milk urea nitrogen breeding value group (high or low), diet (RG or PL), and MUNBV group × diet interactions were used as fixed effects for all analysis. Day (1 and 12 (first and last day of the experiment)) was used as a fixed effect for rumen fermentation characteristics, and its interaction terms with diet and MUNBV groups. The day of treatment in the metabolism crate (1, 2, or 3) was nested within run (1, 2, 3, and 4) and run itself was fitted as a random effects for analyses involving faecal and urinary characteristics, water and N balance, digestibility, purine, microbial efficiency, and milk characteristic analysis. Run + date of the trial was fitted as a random effect for intake analysis with individual cow’s milk production breeding value fitted as a covariate. The milk production breeding value was found to have no effect and was thus removed from the final model. Run was included as a random effect for VFA analysis, and the date of the trial was included as a random effect for botanical analysis. Time of day (am or pm) nested within date was used as the random effect for herbage quality analysis. Herbage chemical composition values were analysed for every day animals were housed within metabolism crates and then reported as an average value for the duration of the trial. Statistical significance was determined at *p* ≤ 0.05 and tendencies are discussed at 0.05 < *p* ≤ 0.10. All statistical analyses were conducted using R [48].

## 3. Results

### 3.1. Herbage Measurements 

Ryegrass diets contained 13 ± 6% weed, 4 ± 0.8% dead material, 3 ± 3% reproductive stem, and 80 ± 4% leaf. The PL diet was comprised of 9 ± 5% weed, 2 ± 1% dead material, 17 ± 3% reproductive stem, and 73 ± 5% leaf. The chemical composition of the diets is presented in Table 1. Differences were detected (*p* ≤ 0.01) between the RG and PL diet for all measured chemical and nutritive value entities, except for DMD% (*p* = 0.15). Ryegrass had a 45% greater DM% content compared to PL. RG also had a 4% greater OM% content, 11% greater CP% content, 70% greater NDF% content, 9% greater ADF% content, 2% greater OMD% content, and 4% greater ME (MJ/kg DM) content, whilst PL had a 19% greater WSC% content.

### 3.2. Herbage Intake and Digestibility

No main effects of MUNBV or diet or diet by MUNBV interaction were detected for either herbage DMI or energy (ME, MJ/kg DM/d) intake (*p* > 0.05), with cows consuming, on average, 15.38 ± 1.13 kg DM/day and an average ME of 181 ± 8.6 MJ/d. Only an effect of diet (*p* < 0.05) was detected for N intake (g/d), with cows on the RG diet consuming on average 384.7 g N/d, which was a 15% greater N intake than cows consuming PL (333.0 g N/d). Only dietary effects were detected on truly digestible values (Table 2). Cows consuming the RG diet had a greater concentration (*p* < 0.05) of truly digestible values for DM (6%), OM (10%), N (12%), ADF (35%), and NDF (69%).

### 3.3. Milk Production

Table 3 presents the milk production data. A tendency for a MUNBV by diet effect (*p* = 0.06) was seen, with low MUNBV cows on RG having higher milk protein percentages. A tendency (*p* = 0.06) was only detected for MUNBV on milk protein yield, with low MUNBV cows tending to have a greater yield (kg/d) than high MUNBV cows. No main effect or interaction effects (*p* > 0.05) were detected for either milk (kg/d) or fat yield and fat percentage. Lactose yield (kg/d) was 9% greater for low MUNBV cows compared with high MUNBV cows (*p* < 0.01), with no effect detected for diet or the MUNBV by diet interaction (*p* > 0.05). A tendency (*p* = 0.09) was only detected for low MUNBV cows to have greater values for milk solids (fat + protein content, kg/d). A main effect of MUNBV (*p* < 0.01) was detected, with low MUNBV cows having a 7% increase in total solids (total production-water, kg/d) compared with high MUNBV cows. Only diet (*p* < 0.01) and MUNBV effect (*p* < 0.01) were detected for daily MUN (mg/dl), with cows on the PL diet having 30% less MUN (mg/dL) compared with the RG diet, whilst low MUNBV cows had a 15% decrease in MUN (mg/dL) compared with high MUNBV cows.

### 3.4. Rumen

The rumen characteristics are presented in Table 4. Glucogenic VFAs decreased 18% from day 1 to day 12 (*p* < 0.01) whilst non-glucogenic VFAs increased 5%. During the same period, the acetate to propionate ratio also increased by 26%. No main or interaction effects (*p* > 0.05) for NH_3_ or total VFA concentrations were detected. 

### 3.5. Purine Excretion and Microbial Crude Protein Flow and Efficiency

Table 5 presents the purine derivatives, creatinine concentrations, and microbial N dynamics. A main effect of diet (*p* < 0.01) was detected, with cows fed ryegrass having a 43% greater concentration of creatinine (mmol/L), 75% greater concentration of hippuric acid (mmol/L), 55% greater concentration of allantoin (mmol/L), and 54% greater concentration of total purine derivatives (mmol/L) compared to cows fed PL.

An interaction between MUNBV and diet was detected (*p* = 0.04) for microbial crude protein flow (g/d), with low MUN cows on the RG diet having an 18% increase in flow compared with all other treatments. Neither diet, MUNBV, or the interaction (*p* > 0.05) affected the microbial efficiency (g of MCP/kg TDOM).

### 3.6. Excretory Pattern

Table 6 presents the excretion characteristics for urine and faeces. A MUNBV by diet interaction (*p* < 0.01) was detected for UUN g/L, resulting in high MUNBV cows consuming the RG diet having a 31% greater concentration than low MUNBV cows on RG. High and low MUNBV cows consuming RG had a 143% and 85% increased UUN g/L concentration compared with both high and low MUNBV cows consuming PL, respectively. Cows consuming PL had no difference in UUN g/L concentrations based on MUNBV (*p* > 0.05). A MUNBV by diet interaction (*p* < 0.01) was also detected for UN g/L, resulting in high MUNBV cows consuming the RG diet having a 29% greater concentration than their low MUNBV counterparts consuming the same diet. High MUNBV cows consuming RG and low MUNBV cows consuming RG were found to have a 123% and 73% greater concentration of UN g/L (*p* < 0.05), respectively, than cows on the PL diet. No difference was detected (*p* > 0.05) in the UN g/L concentration for cows on the PL diet regardless of MUNBV. 

A MUNBV by diet interaction (*p* < 0.01) was detected for UUN, g/event, with high MUNBV cows consuming RG having a 38% greater concentration per urination event compared with low MUNBV cows on the same diet. A 165% and 92% increase (*p* < 0.05) in UUN, g/event was detected for high MUNBV and low MUNBV cows (respectively) on the RG diet compared with cows on the PL diet. No difference was detected between MUNBV groups consuming the PL diet (*p* > 0.05). Figure 1 presents the phenotypical relationship between MUN (mg/dL) and UUN g/event.

A relationship was detected, with g of UUN per event increasing 0.37 g per increase in MUN (mg/dL) across both diets. Diet shifted the intercept, with cows consuming RG having a higher intercept (4.26) compared to cows consuming PL (−0.27). An interaction between MUNBV and diet for UN, g/event (*p* < 0.01) was also detected. High MUNBV cows had a 39% greater UN excretion per event compared with low MUNBV cows on the RG diet. Both high and low MUNBV cows had a 145% and 77% greater UN excretion per event compared to cows consuming PL, respectively. No difference was detected (*p* > 0.05) between high and low MUNBV cows on the PL diet for UN excretions per urination event. 

A main effect of both MUNBV and diet was detected (*p* < 0.01) on urine volume per event (L). Cows consuming RG had, on average, a 7% increase in urine volume/event compared with cows on a PL diet, whilst high MUNBV cows had a 7% increase in urine volume/event compared with low MUNBV cows. An interaction was found between MUNBV and diet (*p* < 0.01) for the number of urination events per day. High and low MUNBV cows consuming PL had no difference in urination events per day; however, on average, cows consuming PL had 3.45 and 6.05 more urination events per day compared to low and high MUNBV cows consuming RG, respectively. An interaction between MUNBV and diet (*p* < 0.05) was also detected for total daily urine volume (L/d). Low and high MUNBV cows consuming RG had a 22% and 34% reduction in daily urine volume compared to cows consuming PL, respectively. Only dietary effects (*p* < 0.01) were detected for both UUN, g/d and UN, g/d, with cows consuming RG having a 56% and 36% increase in UUN and UN g/d, respectively, compared to cows consuming PL. A diet by MUNBV interaction was detected for UUN as a percentage of total UN. High MUNBV cows fed RG had a 2% greater content of UUN as a percentage of total UN compared to low MUNBV cows on the same diet. Cows fed PL had a 12% and 10% lower content of UUN as a percentage of UN compared to high and low MUNBV cows fed RG, respectively.

A main effect of MUNBV and diet was detected for urine patch area, with high MUNBV cows having a 7% greater area per urine patch compared to low MUNBV cows. Cows on the RG diet also had a 7% greater area per urine patch compared to cows on the PL diet. An interaction effect was found between diet and MUNBV for UUN loading rate, with high MUNBV cows on RG diet having a 39% greater loading rate than low MUNBV cows on RG. High and low MUNBV cows on RG had a 164% and 90% greater loading rate (respectively) compared to cows on the PL diet, which were not considered different based on MUNBV.

A MUNBV by diet interaction was detected (*p* < 0.01) for the amount of N (g) per faecal event. Low MUNBV cows offered RG had a 41% increase in N content compared with high MUNBV cows on the same diet. Low MUNBV cows fed RG had a 22% increase whilst high MUNBV cows fed RG had a 14% decrease in N content per faecal event compared to cows of both high and low MUNBV offered PL. An interaction term was also detected for MUNBV by diet (*p* < 0.01) for OM content per faecal event. Low MUNBV cows offered RG had 45% more OM in faeces than high MUNBV cows on the same diet and were 23% and 11% higher than low and high MUNBV cows offered the PL diet, respectively. An interaction term was also detected for MUNBV by diet (*p* < 0.01) for ADF content per faecal event, with high MUNBV cows offered the PL diet having 8% more ADF in the faeces than low MUNBV cows offered the same diet and 75% and 23% more than high and low MUNBV cows on the RG diet, respectively. A MUNBV by diet interaction term was also detected (*p* < 0.01) for NDF content per faecal event, with high MUNBV cows offered PL having 56% more NDF than high MUNBV cows offered RG. High MUNBV cows on RG had an 8% greater NDF content per faecal event than low MUNBV cows irrespective of diet. High MUNBV cows on RG had a 19% lower content of NDF per faecal event compared to low MUNBV cows irrespective of diet. An interaction (*p* < 0.01) between MUNBV and diet was detected for the DM content (g) per faecal event. Low MUNBV cows offered the RG diet had 45% more DM/event than high MUNBV cows offered the same diet and 31% and 18% more DM/event than low and high MUNBV cows offered the PL diet, respectively. A diet by MUNBV interaction (*p* = 0.03) was also detected for the number of faecal events per day. Low MUNBV cows offered RG had 29% fewer faecal events than high MUNBV cows offered both diets and low MUNBV cows on the PL diet, all of which did not differ (*p* > 0.05). Only a diet effect was detected (*p* < 0.01) for total daily faecal N content (g) and total daily faecal DM content (kg), with cows offered the PL diet having a 15% greater N content and 14% greater total DM content per day compared with cows on the RG diet.

### 3.7. Water Balance

Table 7 presents the average daily water balance. Main effects of MUNBV (*p* = 0.03) and diet (*p* < 0.01) were detected on daily water consumed from the trough. Low MUNBV cows drank 44% more water than high MUNBV cows, whilst cows on the RG diet consumed 6 times more water than cows on the PL diet. A dietary effect was also detected for daily water consumed in feed (*p* < 0.01), with cows on the PL diet consuming 58% more water in the forage than cows on the RG diet. No effect of MUNBV or interaction between MUNBV and diet were detected for total water consumption per day. However, a tendency (*p* = 0.06) was detected for diet, with cows consuming PL having larger volumes of water intake than cows consuming RG. An interaction between MUNBV and diet (*p* < 0.05) resulted in less total daily water excretion via urine from high MUNBV cows consuming RG compared to low MUNBV cows consuming RG, whilst cows consuming PL had the greatest volume of water excreted. An effect of diet (*p* < 0.01) was detected for water excreted in faeces, with cows on the RG diet having 16% greater volume of water excreted than cows on PL. A tendency (*p* = 0.09) was detected for MUNBV to have an effect, with high MUNBV cows having larger volumes of water excreted in their faeces. No effect of diet, MUNBV, or their interaction term was detected for the volume of water excreted in milk (*p* > 0.05). Only a dietary effect was detected (*p* < 0.01) for total water excreted, with cows on PL having a 10% greater volume of total water excretion than cows on RG diet. No diet, MUNBV, or interaction effects were detected for overall water balance (*p* > 0.05). A MUNBV (*p* = 0.02) and diet (*p* < 0.01) effect was detected for the amount of water absorbed (%) as a proportion of intake. Cows with low MUNBV had 5% more water absorbed as a proportion of intake compared with high MUNBV cows whilst cows on PL had 11% more water absorbed as a proportion of intake than cows on RG.

### 3.8. Nitrogen Balance and Partitioning

Table 8 presents the N balance and N partitioning. An effect of diet was detected (*p* < 0.05) for differential N partitioning for milk, urine, and faeces. Cows consuming PL partitioned more N into milk (13%) and faeces (30%) but less into the urine (14%) than cows consuming RG. A diet effect was detected for the total amount of N excreted, with cows consuming PL excreting 9% less N compared to cows consuming RG. No main effects of MUN or diet or an interaction term (*p* > 0.05) were detected for the N balance.

## 4. Discussion

We investigated whether cows divergent for MUNBV had a difference in their N balance and if there were any dietary effects or interaction between diet and MUNBV, with the hypothesis that cows divergent for MUNBV would have differences in their N balance and an additive effect of diet and MUNBV on reducing UUN excretions. Our results indicate that there was no difference in the total amount of N excreted based on MUNBV, which is consistent with the recent work of Müller et al. (2021) [49]. There was an effect of diet on total daily N excreted but no interaction with MUNBV. However, our results do indicate that the excretion patterns and characteristics of excreta differ according to MUNBV and diet, which can result in different environmental impacts.

Only dietary effects were detected for the daily N balance, with cows fed the PL diet having a lower amount of total N excreted. This difference can largely be accounted for as the cows on the PL diet had a lower N intake compared with RG, thus resulting in less N excreted. The low content of CP in the RG diet may account for the low N excretion rates from RG compared to the literature [50]. Differential N partitioning from the PL diet was observed, with cows excreting more N through faeces than urine compared with RG cows, which has been reported previously [51,52]. The increased N partitioning to faeces by PL is likely a result of the different plant secondary compounds (aucubin, actoside, catalpol) found in PL compared to RG, which has been reported to influence N partitioning in other studies [51]. A study investigating the use of tannin-containing hay showed an increased partitioning of N into faeces rather than urine [53] similar to what was observed in this study. Increased N partitioning to faeces represents an environmental advantage for PL diets, as N in faeces is less at risk of leaching or volatilising to the environment [54,55]. Different partitioning of N was also observed in the urine, with RG cows having greater concentrations of purine derivatives. The differences in purine derivatives can largely be explained by the different digestibility, energy, and CP content of the diets [56]. The interaction between MUNBV and RG for UUN as a percent UN also indicates a difference in N partitioning within the urine, with high MUNBV cows having a greater level of UUN as a percent of UN when consuming RG. This relationship may warrant further investigation.

A difference in lactose yield was detected, with low MUNBV cows having a greater lactose yield than high MUNBV animals. A similar relationship was reported by Marshall et al. (2020) [32]. Whilst the New Zealand dairy industry does not reward lactose yield in milk payments, there has been growing interest internationally in the refinement of lactose from milk products for use as a component in pharmaceutical products and lactose-fortified food products [57,58]. This could theoretically offer an economic incentive for the use of low MUNBV cows in markets where milk producers pay for lactose content.

Although not different for total daily N excreted, low MUNBV cows consistently had lower concentrations of both UUN and UN when consuming ryegrass, resulting in lower values on a per urination event basis compared with high MUNBV cows. This paired with lower urine volumes per event results in less N being loaded per urine patch for low MUNBV cows consuming ryegrass. The urine patch is considered the “engine room” of N cycling in grazed pastures as it ultimately determines the amount of N deposited to pasture, which can then either be used by plants or lost to the environment [7]. It is possible to estimate the amount of potential nitrate leached based on the urine N loading rates at the urine patch using the empirical model of Di and Cameron (2007) [59] (5):(5)$$y=16.7+0.173x+0.000071x^{2}$$
where y = total nitrate nitrogen (NO_3_N) leached (kg N ${\mathrm{ha}}^{-1}$) and x = UUN load onto pasture (kg/${\mathrm{ha}}^{-1}$). A 1% difference exists between the estimated leaching rate of high and low MUNBV cows consuming PL per urine patch. Low MUNBV cows consuming RG are estimated to have a 23% reduction in nitrate leaching compared to high MUNBV cows consuming RG. Cows consuming PL regardless of MUNBV are estimated to have a 32% and 47% reduction in nitrate leaching compared to low and high MUNBV cows consuming RG, respectively. These reductions in estimated nitrate leaching from the urine patch indicate the ability for both animal genetics with regards to MUNBV when grazing RG diets and the use of PL to reduce the amount of N lost from leaching events. Reductions in N leaching are considered a key objective for reducing eutrophication [60] and therefore meeting legislative concerns nationally and internationally regarding freshwater quality and for reducing any possible negative human health impacts from high N concentrations in drinking water. Not only will the environmental impact on waterways be reduced through the use of MUNBV on RG diets and the use of PL, but the ability to reduce UUN content in the urine will also help reduce GHG emissions. A lower content of UUN per urine patch would be expected to yield comparatively lower amounts of N_2_O emissions [61] and thus reduce GHG emissions from low MUNBV cows grazing RG or cows grazing PL.

The tendency for low MUNBV cows and the effect of the PL diet to have more urination events means that N deposited in urine is likely to be spread over a larger spatial area over a longer time period, which could reduce the N loading in any one area. However, on a per hectare basis, this may result in greater overlapping of urine patches. The effectiveness of low MUNBV cows and the PL diet to reduce N leaching should be further assessed on a per hectare basis. The use of PL should also be assessed from an animal welfare perspective in regard to potential collateral long-term effects of diuresis.

The lack of effect from MUNBV on total daily N excretion from urine despite the consistently lower N concentration on RG diets from low MUNBV cows may be explained by the greater daily urine volume from low MUNBV cows on RG. A greater volume of excretion with a lower N concentration from low MUNBV cows on RG can equate to a similar level of N excretion of a smaller volume but a higher N concentration from high MUNBV cows on RG.

A similar relationship as to that seen with urination characteristics and total N balance was observed in faecal characteristics. Although there was no difference in total daily N excreted in faeces by MUNBV, the behavioural characteristics of the faecal events differed. A greater partitioning of N to faeces from low MUNBV cows consuming RG was observed on a per-event basis, which has associated positive environmental benefits as mentioned earlier when discussing the PL effect of N in total daily faeces. The differences caused by the diet faction of the diet by MUNBV interaction can largely be attributed to the different digestibility observed between RG and PL diets; however, no differences were detected in digestibility variables by MUNBV. As these differences are ‘seen’ per event, they might therefore suggest different dynamics in rumen digesta outflow based on MUNBV. Thus, this results in a different temporal pattern of excretion of the non-digestible diet components.

The difference in MCP flow between the MUNBV groups supports the idea that these cows could have had different rumen digesta outflow dynamics. Low MUNBV cows fed RG had a greater MCP flow, which would result in a greater content of amino acids of high biological value supplied to the small intestine and which could have increased animal productivity [44,62] as potentially represented by the tendency for greater milk protein yield from low MUNBV cows on RG. Similar results were detected by Souza et al. (2021) [63], who found that cows considered high for MUN had a suppressed rate of urea transport to the gastrointestinal tract compared to low MUN cows. Souza et al. (2021) [63] went on to hypothesize that these high MUN cows may therefore be more susceptible to rumen degradable protein deficiencies than low MUN cows, which may have also occurred in this study.

The increased MCP flow from the low MUNBV cows paired with greater voluntary water intake is indicative of a difference in the rumen osmotic concentration [64]. Osmolality may be increased due to increased rates of fermentation [65] in low MUNBV cows. The latter may either be as a result of different rumen microbial communities, greater oral processing of ingesta, or both. Cows divergent for genetic parameters have been reported to have different oral processing behaviour of ingesta [66]. If low MUNBV cows have increased oral processing, it could result in smaller particle sizes entering the rumen [67], which would increase rumen fermentation rates and digesta outflow flow, potentially accounting for some of the differences observed based on MUNBV. No differences in total VFA concentration were detected; however, no measurements of total VFA pools or absorption were made, thus further research is required in this area.

If cows did have a difference in rumen digesta outflow, it would also help explain differences in urination behaviour based on MUNBV. Greater rumen digesta outflow rates, especially the liquid phase, represent a greater content of available water in the body’s water pool that can be used in respiration, evaporation, milk, faeces, or urine [68]. Cows are stimulated to urinate when the bladder capacity reaches 50% [68]. If there was a greater continual outflow of liquid from the rumen entering the body’s water pool and eventually the bladder, it would theoretically reach this threshold at a higher frequency, resulting in more urination events per day as per what was seen with low MUNBV cows. Additionally, low MUNBV cows may have a lower urea N concentration threshold, in which case the dilution function of the kidney would be exacerbated and ultimately increasing the liquid flow to the bladder.

A lack of diet effect from PL was seen throughout this experiment for cows differing in MUNBV, e.g., no difference was observed in UUN concentrations between high and low MUNBV despite the fact this relationship has been established previously on an RG diet [32]. This may in part be due to the interaction of the MUNBV model and differing diets. A genetic model was used to predict MUNBV values, and the parameters used are based on cows grazing predominantly RG/white clover pastures in New Zealand. There is the potential that these parameters may be different if these cows were grazing alternative diets, such as PL, compared to the typical RG/white clover diet used in New Zealand.

Figure 1 presents the phenotypical relationship between MUN (mg/dL) and UUN g/event. A clear effect of MUN (mg/dL) was observed, where an increasing MUN (mg/dL) content increases UNN g/event. This represents the physiological relationship that is occurring within these cows, which was unable to be captured based on a BV for the PL diet, indicating that the model’s parameters may not be appropriate for the PL diet as the outcome does not match the data generating process [69]. New Zealand dairy pastoral systems are largely RG based [70], which would allow for the current MUNBV models to be used with sufficient accuracy to determine N losses in the typical herd. However, the model parameters should be re-assessed using the parameters of cows grazing multiple different forage species—a clear opportunity to add functionality to the model—before any legislative or management decisions are made based on cow MUNBV values in production systems that are not based on RG-dominated swards.

## 5. Conclusions

No effect of MUNBV was detected for total daily urinary N excretions, indicating that cows divergent for MUNBV will be depositing similar amounts of N to pasture via urine on a total daily basis. However, due to the different urination patterns from low MUNBV cows consuming RG, the environmental impact would be expected to differ. Due to the lower concentration of N per event and the subsequent lower urine N loading rate, it is estimated that low MUNBV cows will have less N leaching from urine patches and will excrete more N as faeces when consuming RG pastures compared to high MUNBV cows, thus potentially reducing the environmental impact. Plantain diets also offer a potential opportunity to reduce the environmental impact of pastoral dairy production systems. The PL diet not only resulted in more partitioning of N into faeces, it also reduced the content of N excreted and therefore potentially leached at the urine patch as well as reducing the total daily N excreted compared to the RG diet for both high and low MUNBV cows. Although a limited effect of breeding value for MUN was detected when cows were fed PL, a phenotypical effect was detected with low MUN (mg/dL), resulting in less UUN g/L and therefore reduced environmental impact. The MUNBV model’s parameters must be re-assessed for different forage species, e.g., PL. Once the diet is taken into account in generating MUNBV values, it could be argued that the PL effect may be additive to the positive effect of low MUNBV cows.

## Figures and Tables

**Figure 1 animals-11-02464-f001:**
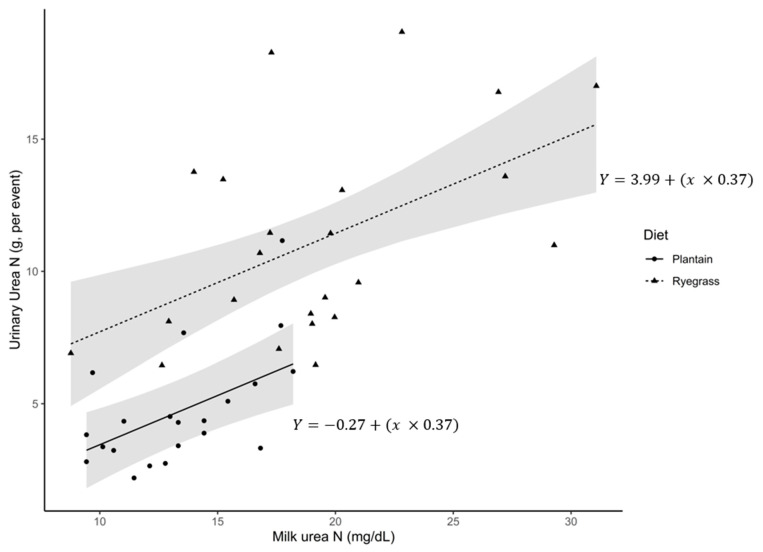
Average daily urinary urea N (UUN) excretion g per event (y-axis) as a function of the average daily milk urea N (MUN, mg/dL) (x-axis) from lactating dairy cows offered either a ryegrass or plantain diet whilst housed in metabolism creates for 72 h. For both diets, a one unit increase in MUN (mg/dL) resulted in a 0.37 ± 0.09 increase in the UUN content (g) per urination event (*p* < 0.01). Diet shifted the intercept of the regression line to 4.26 ± 1.42 for the ryegrass diet and −0.27 ± 1.42 for the plantain diet (*p* < 0.01). The regression model R^2^ = 0.63, with the shaded area representing the 95% confidence interval.

**Table 1 animals-11-02464-t001:** Forage chemical composition of plantain and ryegrass diets offered to cows of differing MUNBVs (milk urea N breeding value) housed in metabolism crates.

	Diet			*p*-Value
Item ^1^	Plantain	Ryegrass	RSD ^2^	Diet
DM, %	12.8	18.5	0.01	<0.01
OM, %	87.7	90.9	0.86	<0.01
CP, %	13.4	14.9	2.00	0.01
WSC, %	19.1	16.0	4.39	0.02
NDF, %	27.3	46.5	2.45	<0.01
ADF, %	24.1	26.2	1.32	<0.01
DMD, %	74.0	75.1	2.52	0.15
OMD, %	77.6	79.5	3.13	0.04
ME, MJ/kg DM	11.1	11.5	0.43	<0.01

^1^ DM, Dry Matter; OM, Organic Matter; CP, Crude Protein; WSC, Water-Soluble Carbohydrates; NDF, Neutral Detergent Fibre; ADF, Acid Detergent Fibre; DMD, Dry Matter Digestibility; OMD, Organic Matter Digestibility; ME, Metabolizable Energy. ^2^ RSD = Residual standard deviation.

**Table 2 animals-11-02464-t002:** Effect of diet (ryegrass and plantain) and MUNBV (milk urea N breeding value) classified as either ‘high’ or ‘low’ on true digestible values for dairy cows housed in metabolism crates.

	High MUNBV		Low MUNBV			*p*-Value		
Item ^1^	Plantain	Ryegrass	Plantain	Ryegrass	RSD ^2^	MUNBV	Diet	MUNBV × Diet
Dry Matter	70	74.5	69.7	74.1	4.45	0.8	<0.01	0.49
Organic Matter	70	76.8	69.5	76.4	3.99	0.72	<0.01	0.5
Nitrogen	64	72	64.2	72.1	4.79	0.92	<0.01	0.39
ADF	54.9	74.3	53.6	72	0.1	0.32	<0.01	0.33
NDF	46	78.4	44.7	74.8	0.13	0.36	<0.01	0.22

^1^ ADF = Acid Detergent Fibre; NDF = Neutral Detergent Fibre. ^2^ RSD = Residual standard deviation.

**Table 3 animals-11-02464-t003:** Effect of diet (ryegrass and plantain) and MUNBV (milk urea N breeding value) classified as either ‘high’ or ‘low’ on daily milk production characteristics for dairy cows housed in metabolism crates.

	High MUNBV		Low MUNBV			*p*-Value		
Item ^1^	Plantain	Ryegrass	Plantain	Ryegrass	RSD ^2^	MUNBV	Diet	MUNBV × Diet
Milk yield, kg	18.6	18.3	19.3	19	1.73	0.15	0.58	0.74
Fat, %	4.54	4.78	4.51	4.75	0.01	0.86	0.19	0.3
Fat yield, kg	0.88	0.84	0.84	0.86	0.01	0.29	0.16	0.27
Protein, %	3.37	3.51	3.41	3.56	0.26	0.45	0.07	0.06
Protein yield, kg	0.62	0.64	0.65	0.67	0.05	0.06	0.37	0.44
Lactose yield, kg	0.93	0.92	1.01	1	0.09	<0.01	0.68	0.76
Milk solids, kg	1.43	1.48	1.49	1.53	0.11	0.09	0.17	0.2
Total Solids, kg	2.49	2.49	2.66	2.66	0.19	<0.01	0.89	0.38
MUN, mg/dl	13.2	19.4	11.8	15.9	0.17	<0.01	<0.01	0.7

^1^ Milk solids = Fat yield + protein yield; Total solids = total production without water; MUN = milk urea nitrogen. ^2^ RSD = Residual standard deviation.

**Table 4 animals-11-02464-t004:** Effect of diet (ryegrass (RG) and plantain (PL)) and MUNBV (milk urea N breeding value) classified as either ‘high’ or ‘low’ on rumen parameter concentrations for dairy cows housed in metabolism crates. There were no significant interaction terms between day or any other factor.

	Day 1				Day 12								
	High MUNBV		Low MUNBV		High MUNBV		Low MUNBV		*p*-Value				
Item ^1^	PL	RG	PL	RG	PL	RG	PL	RG	RSD ^2^	Day	MUNBV	Diet	MUNBV × Diet
Glucogenic	21.3	21.1	20.9	20.8	17.5	17.4	17.2	17.1	0.08	<0.01	0.53	0.79	0.41
Non-Glucogenic	76.3	76.3	76.6	76.6	80	80	80.4	80.4	0.01	<0.01	0.47	0.96	0.45
A:P ratio	3.31	3.34	3.43	3.47	4.13	4.18	4.32	4.38	0.08	<0.01	0.18	0.69	0.35
NH_3_	7.29	6.79	7.16	6.68	5.53	5.24	5.45	5.17	0.44	0.19	0.94	0.75	0.89
Total VFA	73.8	72.6	69.5	68.2	72.3	71.1	67.9	66.7	20.79	0.83	0.56	0.87	0.35

^1^ A:P = Acetate: Propionate ratio; NH_3_ = Ammonia; VFA = Volatile Fatty Acids. ^2^ RSD = Residual standard deviation.

**Table 5 animals-11-02464-t005:** Mean values for purine derivatives and creatinine and microbial nitrogen dynamics for dairy cows classified as either ‘high’ or ‘low’ for a milk urea nitrogen breeding value on either a plantain or ryegrass diet housed in metabolism crates.

Item ^1^	High MUNBV		Low MUNBV			*p*-Value		
	Plantain	Ryegrass	Plantain	Ryegrass	RSD ^2^	MUNBV	Diet	MUNBV × Diet
Purine derivatives (mmol/L)								
Creatinine	1.95	2.81	1.89	2.69	0.22	0.27	<0.01	0.95
Hippuric	22.59	41.08	19.95	33.37	0.35	0.08	<0.01	0.62
Allantoin	4.99	7.81	4.8	7.37	0.24	0.42	<0.01	0.65
Uric	0.39	0.43	0.43	0.48	0.03	0.26	0.34	0.28
Total PD	5.34	8.27	5.26	8.07	0.24	0.52	<0.01	0.72
Microbial N dynamics								
MCP flow, g/d	1058 ^b^	1065 ^b^	1041 ^b^	1240 ^a^	149.23	0.09	0.03	0.04
Microbial efficiency. g of MCP/kg TDOM	96.5	90.3	107	100.7	21.76	0.13	0.36	0.2

^a–b^ Mean values in the same row with different superscript differ (*p* < 0.05) for the interaction between diet and MUNBV group. ^1^ PD = Purine derivatives (sum Allantoin + Uric acid); MCP = Microbial Crude Protein; TDOM = Truly digestible organic matter. ^2^ RSD = Residual standard deviation.

**Table 6 animals-11-02464-t006:** Mean values for urinary and faecal characteristics for dairy cows classified as either ‘high’ or ‘low’ for a milk urea nitrogen breeding value on either a plantain or ryegrass diet housed in metabolism crates.

Item ^1^	High MUNBV		Low MUNBV			*p*-Value		
	Plantain	Ryegrass	Plantain	Ryegrass	RSD ^2^	MUNBV	Diet	MUNBV × Diet
Urination characteristics								
UUN, g/L	1.69 ^c^	4.31 ^a^	1.86 ^c^	3.29 ^b^	0.38	<0.01	<0.01	<0.01
UN, g/L	2.50 ^c^	5.79 ^a^	2.70 ^c^	4.49 ^b^	0.32	<0.01	<0.01	<0.01
UUN, g/event	4.53 ^c^	11.90 ^a^	4.44 ^c^	8.62 ^b^	0.47	<0.01	<0.01	<0.01
UN, g/event	6.71 ^c^	16.20 ^a^	6.50 ^c^	11.66 ^b^	0.04	<0.01	<0.01	<0.01
Volume/event, L	2.63	2.83	2.47	2.65	0.34	0.01	<0.01	0.16
Events/day	15.1 ^a^	9.1 ^c^	15.2 ^a^	11.7 ^b^	0.22	0.06	<0.01	<0.01
Volume/day, L	40.4 ^a^	26.4 ^c^	39.3 ^a^	31.0 ^b^	0.23	0.38	<0.01	0.05
UUN, g/d	67.5	109.12	68.82	102.04	0.23	0.17	<0.01	0.26
UN, g/d	108.1	147.83	103.76	139.85	0.21	0.23	<0.01	0.35
UUN, % UN	65.7 ^c^	75.3 ^a^	66.5 ^c^	73.5 ^b^	4.42	0.17	<0.01	<0.01
Urine patch area, m^2^	0.53	0.57	0.5	0.53	0.34	0.02	0.01	0.15
UUN loading rate, kg UUN ha^−1^	90.0 ^c^	241.1 ^a^	92.3 ^c^	173.7 ^b^	44.45	<0.01	<0.01	<0.01
Faecal characteristics								
N, g/event	6.65 ^b^	5.51 ^c^	6.10 ^b^	7.75 ^a^	0.42	<0.01	0.19	<0.01
OM, g/event	238.65 ^b^	183.10 ^d^	216.28 ^c^	265.22 ^a^	0.39	<0.01	0.05	<0.01
ADF, g/event	105.47 ^a^	60.14 ^d^	97.72 ^b^	85.66 ^c^	0.39	<0.01	<0.01	<0.01
NDF, g/event	144.07 ^a^	92.23 ^c^	133.44 ^b^	133.63 ^b^	0.4	<0.01	<0.01	<0.01
DM, g/event	272.11 ^b^	221.35 ^d^	243.75 ^c^	320.07 ^a^	0.4	<0.01	0.12	<0.01
Events/day	17.65 ^a^	17.81 ^a^	17.92 ^a^	12.71 ^b^	0.23	0.04	0.04	0.03
Total Faecal N/d	117.66	102.5	114.69	100.24	0.18	0.57	0.01	0.47
Total daily, kg/d	4.7	4.12	4.54	3.99	0.01	0.37	<0.01	0.12

^a–c^ Mean values in the same row with different superscript differ (*p* < 0.05) for the interaction between diet and MUNBV group. ^1^ UUN, Urinary Urea Nitrogen; UN, Urinary Nitrogen; N, Nitrogen; OM, Organic Matter; ADF, Acid Detergent Fibre; NDF, Neutral Detergent Fibre; DM, Dry Matter. ^2^ RSD = Residual standard deviation.

**Table 7 animals-11-02464-t007:** Average daily water balance for dairy cows on either a ryegrass or plantain diet classified as either ‘high’ or ‘low’ for MUNBV (milk urea N breeding value) housed in metabolism crates.

	High MUNBV		Low MUNBV		*p*-Value			
Item	Plantain	Ryegrass	Plantain	Ryegrass	RSD ^1^	MUNBV	Diet	MUNBV × Diet
Water consumed, L/d								
Trough	0.8	30.3	9.4	35.5	9.83	0.03	<0.01	0.21
Forage	100.5	62.8	94.9	60.6	0.12	0.12	<0.01	0.87
Total	101.3	93.1	104.3	96.1	13.46	0.47	0.06	0.33
Water excreted, L/d								
Urine	40.4 ^a^	26.4 ^c^	39.3 ^a^	31.0 ^b^	0.23	0.38	<0.01	0.05
Faeces	29.8	34.4	27.5	32.2	4.28	0.09	<0.01	0.88
Milk	16.1	15.7	16.7	16.3	1.56	0.3	0.45	0.76
Total	87.4	77.6	85.2	79.2	0.1	0.83	<0.01	0.28
Water balance	13.9	15.5	19.1	16.9	8.19	0.34	0.36	0.63
Water absorbed, %								
Prop. of intake	70.1	62.9	73.5	66.3	4.5	0.02	<0.01	0.29

^a–c^ Mean values in the same row with different superscript differ (*p* < 0.05) for the interaction between diet and MUNBV group. ^1^ RSD = Residual standard deviation.

**Table 8 animals-11-02464-t008:** Average daily intake and nitrogen balance for cows on either a plantain or ryegrass diet and classified as either ‘high’ or ‘low’ for MUN (milk urea N breeding value) housed in metabolism crates.

	High MUNBV		Low MUNBV		*p*-Value			
Item ^1^	Plantain	Ryegrass	Plantain	Ryegrass	RSD ^2^	MUNBV	Diet	MUNBV × Diet
Intake								
DMI, kg/d	15.5	15.8	14.9	15.3	0.14	0.41	0.57	0.96
ME, MJ/d	179	190	172	183	1.92	0.41	0.19	0.95
N, g/d	338.5	390.1	327.6	379.2	56.34	0.25	<0.01	0.59
Nitrogen excreted								
Milk, % of intake	30.1	26.8	32.9	29	0.2	0.15	0.04	0.69
Urine, % of intake	33.1	38.4	31.5	36.4	0.26	0.4	0.02	0.32
Faeces, % of intake	35.4	27.3	35.8	27.5	0.2	0.85	<0.01	0.55
Total N excreted, g/d	341.4	374.2	329.5	362.8	36.05	0.47	<0.01	0.32
N Balance, g/d	−2.9	15.9	−1.9	16.4	53.5	0.71	0.63	0.96

^1^ DMI, Dry Matter Intake; ME, Metabolizable Energy; N, Nitrogen. ^2^ RSD = Residual standard deviation.

## Data Availability

Not applicable.

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
