# Peer review of "Nitrogen Balance of Dairy Cows Divergent for Milk Urea Nitrogen Breeding Values Consuming Either Plantain or Perennial Ryegrass"

_animals, 2021, doi:10.3390/ani11082464_

Round 1
Reviewer 1 Report
This is a very interesting study on the effect of milk urea nitrogen breeding value and forage-based diet (i.e. perennial ryegrass and plantain) on urination behavior, N partitioning and on environment. It is written and discussed well but there are still a few points to address before it is accepted. I am only not sure about the number of animals used in the study. It appears to me very low. Please see below for the specific comments.
Ln 12. Simple summary looks like to be a second abstract. It should be written more succinctly.
Ln 54. I would recommend authors to cite Oudshoorn (2008) here instead #5 as this is the thesis work is broader than the scope of the particular study.
- Oudshoorn, F. W., Kristensen, T., & Nadimi, E. S. (2008). Dairy cow defecation and urination frequency and spatial distribution in relation to time-limited grazing. Livestock Science, 113(1), 62-73.
Ln 57. This is not only a particular problem of New Zealand dairy farming. I would recommend author to indicate that as they are publishing in an international journal.
Ln 65. It appears that only the authors of the current manuscript studied those dietary and management strategies! Too much self-citing including a conference proceeding. See some relevant studies on this as well:
- Bryant, R. H., Dalley, D. E., Gibbs, J., & Edwards, G. R. (2014). Effect of grazing management on herbage protein concentration, milk production and nitrogen excretion of dairy cows in mid‐lactation. Grass and Forage Science, 69(4), 644-654.
- Dodd, M., Dalley, D., Wims, C., Elliott, D., & Griffin, A. (2019). A comparison of temperate pasture species mixtures selected to increase dairy cow production and reduce urinary nitrogen excretion. New Zealand Journal of Agricultural Research, 62(4), 504-527.
- Carmona-Flores, L., Bionaz, M., Downing, T., Sahin, M., Cheng, L., & Ates, S. (2020). Milk production, N partitioning, and methane emissions in dairy cows grazing mixed or spatially separated simple and diverse pastures. Animals, 10(8), 1301.
- Thomson, B. C., Ward, K., Smith, N., Gibbs, J., & Muir, P. D. (2021). Effect of feeding time on urinary and faecal nitrogen excretion patterns in sheep. New Zealand Journal of Agricultural Research, 1-6.
Ln 67. Although Totty et al (2013) is a highly cited paper, I don’t think it is the most appropriate paper to cite in this context. In Totty et al. (2013), authors compared simple vs. diverse pastures containing a number of forb species including plantain. I would recommend authors to consider citing Box et al., (2017) and Bryant et al., (2020). I am sure the authors must be aware of this excellent review paper on this topic.
- Box, L. A., Edwards, G. R., & Bryant, R. H. (2017). Milk production and urinary nitrogen excretion of dairy cows grazing plantain in early and late lactation. New Zealand Journal of Agricultural Research, 60(4), 470-482.
- Bryant, R. H., Snow, V. O., Shorten, P. R., & Welten, B. G. (2020). Can alternative forages substantially reduce N leaching? findings from a review and associated modelling. New Zealand Journal of Agricultural Research, 63(1), 3-28.
I cannot check every individual reference. Please make sure that you are citing contextually most appropriate studies.
Ln 72-74. Break this sentence into two for better flow.
Ln 76 add a comma after in a grazing setting,
Ln 84. English is not my mother tongue but I believe this should be “We hypothesise (British English) or we hypothesize (American English). Please check the manuscript thoroughly based on the language requirement of the journal.
Ln 88 and 89. Based on your botanical composition, the diets were not 100% plantain or perennial ryegrass. I suggest you to modify this to “pure” or “monoculture”.
Ln 94. Check the geographical location (-43.65 º North, 172.33 º East). Something is off here. Note that the fractional portion of decimal degrees are between 0 and 60 (can’t be 65). In another study (Fleming et al 2018), the same location was indicated as “Ashley Dene Research and Development Station in Canterbury”. Are the authors referring to the same facility or two different facilities in the same location? Google earth© says Johnston Memorial Laboratory is located at (43°38′ S, 172°27′ E, and 13 m above sea level).
Ln 109. I am not sure but the number of cows per group seems to be low. Although this was a metabolism study in crates, what was the justification for the low number of cows? Did the authors run any power test?
Ln 114 what was the average milk yield for each treatment?
Ln 150 were cows were monitored continuously during 72 hours without any intermittence?
Ln 152 Please provide more details about the urine harness (i.e. model, location of production) and how they were attached to the vulva. Did the authors use catheters? Also, I was wondering why the authors did not use PEETER Urine sensors as they did in their previous study? (Marshall, C. J., Beck, M. R., Garrett, K., Barrell, G. K., Al-Marashdeh, O., & Gregorini, P. (2020). Grazing dairy cows with low milk urea nitrogen breeding values excrete less urinary urea nitrogen. Science of The Total Environment, 739, 139994.) Was there any specific reason or any advantage to choose urine harnesses? In anyway, more details are needed on the urine harnesses.
Ln 156 acidified to what pH level? What was the concentration of sulfuric acid?
Ln 171 spell out the Conc.
Ln 180 spell out the PD.
Ln 222. How were the herbage measurement analyzed? As far as I understand, there was no replication. You need to provide the details in 2.3. Statistical Analysis section.
Ln 261. Where is the data for herbage intake and digestibility? The results were only mentioned in the text. Why?
Ln 276 what is water yield?
Ln 291 Where is the data? I don’t understand why the author refrained themselves presenting the data even though no differences were detected.
Ln 438. Overall, the section is written well but there are a few interesting points still were not discussed. Authors should also discuss the difference in lactose yield.
Ln 452. The role of hydrolysable tannins found in plantain should be discussed here. The authors have not mentioned anything about the role of plant secondary metabolites in N partitioning. See
- Stewart, E. K., Beauchemin, K. A., Dai, X., MacAdam, J. W., Christensen, R. G., & Villalba, J. J. (2019). Effect of tannin-containing hays on enteric methane emissions and nitrogen partitioning in beef cattle. Journal of Animal Science, 97(8), 3286-3299.
Author Response
I have attached a word document with my responses to all of your comments.

Reviewer 2 Report
Comments and Suggestions for Authors
The manuscript entitled "Nitrogen Balance of Dairy Cows Divergent for Milk Urea Nitrogen Consuming Either Plantain or Perennial Ryegrass" is focused on the evaluation of N balance of lactating dairy cows selected for milk urea N concentration breeding values fed on two different forages from the view of environmental aspects. In my opinion, the experimental design and the parameters analysed were chosen properly to answer the aims of the study. The results are interesting and the work is a significant contribution to the field so it is worth publishing in Animals, however, there are some points that should be addressed.
Introduction
Please add information about bioactive compounds in plantain (Plantago lanceolata) because they have diuretic (iridoid glycosides) and antimicrobial and antifungal effects (aucubin and acteoside) and can influence N metabolism in the rumen (reduction of NH3) and N excretion. These aspects should not be omitted in the background information for the study.
Material and Methods
L. 156 - Please add concentration and amount of sulphuric acid used for urine acidification.
Results
L. 290 – Can you add information about the effect of MUNBV and diet on the A/P ratio?
L. 328 – 330 – It will be better to express the relationships between milk urea nitrogen and urinary urea nitrogen for both diets in the form of equations instead of verbal description or you can add the equations into the Figure 1.
Figure 1 – Was the R2 value same for both diets?
Table 5 – Is the consumption of water from trough in High MUNBV – Plantain of 0.8 L/d correct? It is unbelievably low. How was calculated water absorbed as a proportion of intake? Is it in L/d? Expressing this value in % could be better.
Discussion
L. 452 – 455 – What is the mechanism of a shift in a way of N- excretion?
Please can you also discuss a little bit perspectives/possible risks of long time feeding of plantain to ruminants from the view of animal health (diuretic effect)?
References
Please check the references for formal style (e.g. ref. no 24 – title in italics, journal name in full).
Author Response
Thank you for your comments, i have attached a word document with responses to all of your comments

Round 2
Reviewer 2 Report
The manuscript titled " Nitrogen balance of dairy cows divergent for milk urea nitrogen consuming either plantain or perennial ryegrass" has been re-evaluated.
The authors have taken into consideration my suggestions and revised the manuscript accordingly. In my opinion it is now suitable for publication.
Thank you.
Author Response
Thank you for all the revisions you have provided.